# Evaluation of the Effectiveness of Active Vitamin D Use in Experimental Rat Lymphedema Model

**DOI:** 10.3390/medicina60111788

**Published:** 2024-11-01

**Authors:** Dicle Aksöyler, Erol Kozanoğlu, Mehmet Korkut, Ömer Faruk Akpınar, Muhammet Çaylı, Hüseyin Can Yücel, Bora Akalın, İlkay Özdemir, Fatma Beyza Sağ, Seyhun Solakoğlu, Alpaslan Mayadağlı, Rıfat Atılla Arıncı, Luigi Losco

**Affiliations:** 1Department of Plastic Reconstructive and Aesthetic Surgery, Istanbul Faculty of Medicine, Istanbul University, Istanbul 34093, Turkey; erol.kozanoglu@istanbul.edu.tr (E.K.); krktmehmet94@gmail.com (M.K.); farukomerakpinar@gmail.com (Ö.F.A.); mcayli98@gmail.com (M.Ç.); h.can3132@gmail.com (H.C.Y.); akalinbe@gmail.com (B.A.); aarinci@istanbul.edu.tr (R.A.A.); 2Graduate School of Health Sciences, Istanbul Medipol University, Istanbul 34815, Turkey; ilkay.ozdemir@std.medipol.edu.tr; 3Department of Histology and Embryology, Istanbul Faculty of Medicine, Istanbul University, Istanbul 34452, Turkey; beyzasag@istanbul.edu.tr (F.B.S.); seyhunsol@gmail.com (S.S.); 4Department of Radiation Oncology, Bezmialem Vakif University, Istanbul 34093, Turkey; amayadagli@bezmialem.edu.tr; 5Department of Medicine, Surgery and Dentistry, University of Salerno, 84084 Fisciano, Italy; luigi.losco@gmail.com

**Keywords:** active vitamin D, fibrosis, lymphangiogenesis, lymphedema rat model, M1 macrophages, M2 macrophages, M2/M1 macrophage balance

## Abstract

*Background and Objectives*: Lymphedema is a progressive, chronic condition. Traumatic damage to the lymphatics, removal of lymph nodes, and/or radiation are the major causes of fibrosis and a subsequent pathological cascade. Macrophages play a crucial role in wound healing, with M1 macrophages known for their pro-inflammatory effects and M2 macrophages recognized for their anti-inflammatory effects, including improved angiogenesis, lymph angiogenesis, and tissue healing. This study aims to assess the use of calcitriol to alter the M2/M1 macrophage balance, reduce tissue fibrosis in a lymphedema model, promote new micro-lymphatic vessel formation, and evaluate the benefits of active vitamin D. *Material and Methods*: Forty-five rats were randomly divided into three groups: control surgery (group A), surgery with preoperative–postoperative calcitriol (group B), and postoperative calcitriol (group C). One week after the surgical ablation a total dose of 20 Gy radiation therapy was administered to the operated groin region. Micro-computed tomography was used for limb volume calculation, fluorescence lymphatic imaging was used to assess the presence of lymphedema, and histopathological analyses were conducted to evaluate the M1/M2 macrophage ratio, fibrosis accumulation, and lymph angiogenesis. *Results*: The micro-computed tomography evaluation revealed that 75% of the rats of group A exhibited long-lasting lymphedema. In group B, the initial lymphedema ratio was the lowest, affecting only 25% of the rats. After ligating the main vessels, a linear lymphatic microvascular structure was observed in groups B and C. Group B had a significant increase in M2 macrophages and newly formed lymphatic vessels (*p* < 0.05). However, group A showed a significant elevation of M1 macrophages and collagen accumulation (*p* < 0.05) in the surgically treated hind limb. *Conclusions*: Both histological analyses and clinical results reported a relevant influence of calcitriol administration. Among all groups, the most favorable outcomes were seen in group B (prophylaxis group). Hence, calcitriol administration could play a crucial role in enhancing the migration of M2 macrophages to the damaged tissue. Such migration may contribute to lymphedema resolution either by enhancing the organization of superficial lymphatic vessels or resolving fibrosis, or with a combination of both these mechanisms.

## 1. Introduction

Lymphedema is a chronic progressive condition characterized by swelling, fibrotic tissue accumulation, and impaired immune response, occurring secondary to a reduction in the number of lymphatic vessels, dysfunction of lymphatic vessels, or obstruction in lymphatic flow [1,2,3]. Clinical and experimental studies have illustrated the critical relationship between fibrosis development and the progression of lymphedema disease [2,4,5]. Several studies demonstrated that interventions aimed at decreasing tissue fibrosis can lead to significant improvements in lymphatic function and a reduction in the pathological consequences of lymphatic damage in rodent lymphedema models [6,7].

Macrophages are responsible for wound healing, lymphangiogenesis, and fibrosis, and they are divided into two subtypes [5]. Classically activated M1 macrophages primarily promote inflammation, tissue damage, and other proinflammatory effects, while alternatively activated M2 macrophages are recognized for their anti-inflammatory and immunoregulatory functions [8].

Calcitriol (C_27_H_44_O_3_), also known as 1,25-dihydroxy vitamin D_3_, has many functions related to bone and mineral metabolism [9]. Recently, calcitriol has been linked to various anti-inflammatory effects by reducing oxidative stress, suppressing inflammatory reactions, and promoting vascularization via increased VEGF in rodent models of random skin flaps, diabetic glomerulonepropathy models, and some clinical trials [9,10,11,12,13]. Previous studies indicated that calcitriol can influence the M2/M1 macrophage balance within damaged tissue by switching macrophage phenotypes from M1 to M2 through various biochemical pathways [10,14].

The purpose of this study is to assess the macroscopic and histological efficacy of systemic calcitriol use for improving the resolution of lymphedema in a rat lymphedema model. Furthermore, we aim to evaluate the potential use of calcitriol as a prophylactic or therapeutic agent in human lymphedema disease.

## 2. Materials and Methods

### 2.1. Lymphedema Rat Model

Forty-five non-immunosuppressed male 8-month-old Sprague–Dawley rats, weighing between 350 and 400 g, were randomly separated into three groups (*n* = 15 per group). All animal procedures were performed and complied with Animal Research Center of Aziz Sancar Experimental Medicine Research Institute Laboratory Animal Production and Purification Department Guidelines. All rats underwent an experimental secondary lymphedema model by receiving surgical ablation and radiotherapy. The groups were as follows: control group (group A), prophylaxis vitamin D group (group B; preoperative 7 days and postoperative 10 weeks of 1.2 μg/kg per day intraperitoneal calcitriol), and treatment vitamin D group (group C; postoperative 10 weeks of 1.2 μg/kg per day intraperitoneal calcitriol). The calcitriol (Calcijex (1 mcg/1 mL) solution for injection ampoules from Abbott Laboratories (Chicago, IL, USA) was used in this study [15].

Surgical procedures were performed using a microsurgical loupe (Zeiss 3.5, Jena, Germany) on the right hind leg of the rats to ensure standardization. General anesthesia was administered with intraperitoneal injections of 87.5-mg/kg ketamine hydrochloride (Pfizer, New York, NY, USA) and 12.5-mg/kg xylazine (Rompun; Bayer Healthcare, Whippany, NJ, USA) [2], and both hind limbs were shaved and cleaned. A 0.1-mL volume of 10% Evans Blue (Sigma-Aldrich, St. Louis, MO, USA) was applied to the dorsal aspect of the right paw, and the limb was flexed and extended for five minutes to ensure dye flow from distal to proximal.

After staining the inguinal and popliteal lymph nodes and their accompanying lymphatic ducts, a 0.5-cm-wide circumferential strip of skin and subcutaneous tissue was excised from the right hind limb. The stained popliteal lymph node and an inguinal fat pad containing the inguinal lymph node were excised, and the main lymphatic duct was dissected from the neurovascular bundle. The dyed lymphatic vessels that pierced the biceps femoris and reached the main lymphatic vessels were ligated and cauterized with unipolar hand cautery (The Hotsy 2000 Loop Tip Cautery, St. Louis, MO, USA). The remaining skin was sutured above the muscle proximally and distally with a 1-cm gap. One week after the surgical procedure, under general anesthesia with the same agents and doses, a single dose of radiation was applied to a 4 × 4-cm area over the operated groin region at a rate of 1 Gy/min with a total dose of 20 Gy (Varian 2100 EX Linear Accelerator; Medical Imaging Resources, Ann Arbor, Mich, MI, USA). Topical hydrocortisone cream was applied to any radiation-related burns [16,17] (Figure 1, Figure 2 and Figure 3).

### 2.2. Limb Volume Calculation

Micro-computed tomography (micro-CT; NanoSPECT/CT; Sedecal Argus, Algete, Spain) was performed to compare the surgically treated right hind limb with the untreated left hind limb. Four and eight weeks after surgery, under general anesthesia, four rats from each group underwent the radiological examination. An 8-cm trans-axial field of view encompassed both lower extremities. A 15-min scan with 65-kVp energy, an exposure time of 1000 ms, and 123-lA energy was applied. All images were reconstructed with a filtered back project into a two-dimensional image volume with a pixel size of 0.2 mm in coronal, transverse, and axial sections. All images were stored in the Digital Imaging and Communications in Medicine format and evaluated. The lengths and widths of the soft tissue in the tibiofibular ankle were calculated and multiplied to determine the volume of the extremity [2].

### 2.3. Fluorescence Lymphatic Imaging

At the third, sixth, and tenth weeks after surgery, fluorescence lymphography was conducted to differentiate the lymphatic structures of the normal untreated left hind limb from the surgically induced lymphedema in the swollen right hind limb. Rats anesthetized with isoflurane were placed on a heating pad [2], and a 5-μL volume of 1-mg/mL indocyanine green (ICG; Sigma-Aldrich) solution, dissolved in distilled water, was administered subcutaneously to the dorsal aspect of both paws using a 26-gauge needle. Fluorescence images were obtained 15 min after the ICG injection using a near-infrared fluorescence camera system (Photodynamic Eye; Hamamatsu Photonics, Hamamatsu, Japan).

### 2.4. Histological Analysis

All rats were euthanized in the tenth week after surgery with high dose sodium pentobarbital (Pentobarsol, Dechra Veterinary Products, Leawood, KS, USA) (150–200 mg/mL). [9]

Fifteen rats from each group were sampled. Skin samples were harvested from a point 4 mm distal to the surgical wound and were fixed with 4% paraformaldehyde, then were embedded in paraffin. Sections taken from paraffin blocks in 4-µm pieces were stained with hematoxylin-eosin and Masson’s trichrome, and immunohistochemical staining was performed to assess the levels of lymphedema. Images were obtained under an Olympus IX-70 microscope (Olympus, Tokyo, Japan) for hematoxylin and eosin and immunohistochemical staining. After staining, the specimens were examined under 40×, 100×, and 200× magnifications. Immunohistochemical staining was used to determine the *H* score of newly formed lymphatics by staining lymphatic vessel endothelial receptor 1 (LYVE-1; Abcam, Inc., Cambridge, MA, USA Catalog # AB 281587). M1 macrophages were stained with the IL-12 antibody (Anti-IL12A Antibody; Picoband, Boster Biological Technology, Pleasanton CA, USA, Catalog # A00918-1), and M2 macrophages were stained with Arginase-1 antibody (Anti-IL12A Antibody; Abcam, Inc., Cambridge, Massachusetts, Catalog # A00918-1). The dilution ratios for LVYE-1, IL-12, and Arginin-1 antibodies for immunohistochemistry (IHC) were 1:100, 1:200, and 1:200, respectively. Arginin 1 and Il-12 staining sections were evaluated semi-quantitatively. Cell density in the tissue samples was examined by two different histologists and scored between 0 and 3. If there were no stained cells, it was scored as “0”; if there were a small number of stained cells in isolation, it was scored as “1”; if there were stained cells forming small clusters, it was scored as “2”; and if the stained cells formed large clusters, it was scored as “3”. One slide from each subject was stained and ten fields were evaluated on each slide. H-score values were determined by calculating the average scores in the histological scoring formula. H-score is obtained by the following formula: 3x% of strongly staining cells (3 + intensity) + 2x% of moderately staining cells (2 + intensity) + 1x% of weakly staining cells (1 + intensity), in the range of 0–300. [18] The observers were blinded to the lymphatic vessels and macrophages on specimens.

### 2.5. Statistical Analysis

The calculation (d-value) method developed by Cohen was used to calculate the effect size to be used as a reference in determining the sample size [19]. The effect size for the least significant measurement difference was calculated as d = 0.81, using the findings of the study in the literature investigating the new lymphedema model on the rats. [2] In this context, in the analysis of the quantitative difference between the two groups, d = 0.81 was observed for a change difference of approximately 0.9 units (±1.1 standard deviations). It was deemed appropriate to include a total of 45 rats as the sample group using the G-power (version 3.1.9.7, Kiel, Germany) package program with a 95% confidence level (1−α) and 90% test power (1−β).

Descriptive, graphical, and statistical methods were used to assess the normality of the scores obtained from each continuous variable. The Shapiro–Wilk test was used to evaluate the normality of these scores. Comparisons between groups were conducted using the one-way analysis of variance (ANOVA) test, and the Tukey post hoc test was used where the *F* value was significant. Differences between the right and left legs were analyzed using the paired sample *t*-test. The McNemar test was used for comparing qualitative repeated measures, while Fisher’s exact test was used to compare qualitative variables. Statistical significance was set at *p* < 0.05 (two-sided) within a 95% confidence interval. All analyses were performed using the Statistical Package for the Social Sciences (SPSS) software, version 26 (IBM Corp., Armonk, New York, NY, USA).

## 3. Results

### 3.1. Limb Volume Calculation

Limb measurements of the operated and non-operated hind limbs were compared, and a >10% increase in the surgically treated limb confirmed the diagnosis of lymphedema [20]. Acute lymphedema in the right hind limb was macroscopically confirmed the day after lymph node excision. In group A, three out of four rats exhibited sustained lymphedema at Weeks 4 and 8. In group B, two out of four rats presented with lymphedema at Week 4, with one rat demonstrating sustained lymphedema until Week 8. In group C, three out of four rats exhibited lymphedema at Week 4, with chronic lymphedema observed in one rat at Week 8. There were no statistically significant differences among the groups according to micro-CT examination (Figure 2 and Table 1).

### 3.2. Fluorescence Lymphatic Imaging

Fluorescence imaging was used to objectively assess the presence, persistence, and resolution of lymphedema. Four rats from each group underwent this examination at the postoperative third, sixth, and tenth week of the study period. In group A, the right hind limb exhibited a diffuse pattern, consistent with significant lymphedema, with no evident lymphatic drainage from the distal to the proximal part of the extremity and ICG dye accumulation throughout the study (Figure 3). In group B, some rats exhibited a stardust pattern in Week 3 and a splash/linear pattern in Week 10 (Figure 3c). In group C, some rats exhibited splash/linear or diffuse patterns at the end of the study. Nearly normal distinct linear vessel structures extended across the thigh in these groups (Figure 3d).

### 3.3. Histopathological Analyses

Histological results are summarized in Table 2.

#### 3.3.1. Collagen Accumulation

Collagen scoring was performed by grading collagen increase in the dermis on a scale of 0 to 3. A score of 0 indicated no increase, meaning normal collagen fiber staining in the dermis; 1 indicated a small amount of collagen increase; 2 indicated moderate collagen increase; and 3 indicated a large amount of collagen increase, as described in the literature [21].

In all groups, scar tissue formation due to collagen deposition was greater in the right leg than in the left leg (*F* = 3.814; *p* = 0.030). Subgroup analysis showed that this difference was significant between group A and B rats. Rats in group A had higher collagen levels than those in group B (*p* < 0.05). The difference between right and left leg lymphedema collagen mass was significantly lower in rats of group B (*p* < 0.01 and *p* < 0.05; Figure 4).

#### 3.3.2. LYVE-1 Staining

In all groups, LYVE-1 staining levels were higher in the right leg than in the left leg (*p* < 0.001). In the right leg, LYVE-1 staining levels were highest in group B (*p* < 0.05) and lowest in group A (*p* < 0.05). Group comparisons showed that LYVE-1 staining differences between the right and left legs were significantly higher in group B than in group A (*p* < 0.05; Figure 5).

#### 3.3.3. IL-12 Staining

A significant difference was observed in IL-12 staining levels between the groups for the right (*F* = 19.544; *p* < 0.001). Subgroup analysis revealed differences between all subgroups, with groups B and C showing lower IL-12 staining levels than group A (*p* < 0.05; Figure 6).

#### 3.3.4. Arginase-1 Staining

In all groups, Arginase-1 staining levels were higher in the right leg than in the left leg (*p* < 0.001). Statistically significant differences were found between groups for both the right and left legs (R and L, *F* = 3.441 and 3.422; *p* < 0.05). Arginase-1 staining of rats in group A was lower than in group B (*p* < 0.05; Figure 7).

## 4. Discussion

Fibrosis accumulation and scar tissue formation are key factors in the development and progression of chronic lymphedema [22,23]. Irradiation inhibits fibroblast proliferation, leading to scar tissue formation [24]. To model human secondary lymphedema, particularly related to mastectomy and radiotherapy for breast cancer, we used the rat lymphedema model described by Kanter et al. [1]. This model involved circumferential skin and subcutaneous tissue excision and inguinal and popliteal lymphadenectomy, followed by a single dose (45 Gy) of radiotherapy. The radiotherapy aimed to block the rats’ strong superficial lymphatic collateral circulation and observe macrophage subtype responses to fibrosis [2].

Lymphedema progression in humans is not gender dependent. Even though male rats were chosen in this study in accordance with the literature, results could be implemented to female rats [2,16].

Due to concerns about anesthesia-related complications and the cost of procedures, only four rats from each group underwent micro-CT examination and fluorescence lymphatic imaging. These imaging studies confirmed lymphedema in some rats, although no statistically significant differences were observed between the groups.

In fluorescence imaging, a linear vessel structure pattern was observed in groups B and C, despite the absence of both inguinal and popliteal lymph nodes and the ligation of the main lymphatic vessels. This pattern likely represents a compensatory mechanism for lymphatic drainage via new lymphatic vessels, influenced by calcitriol administration. Furthermore, rats in groups A and C exhibited diffuse patterns at Week 3, whereas rats in group B presented a stardust pattern at Week 3, which involved a splash/linear pattern by Week 10 [25].

Macrophages, key regulators of wound healing, are divided into two subtypes with responses to inflammation similar to those observed in humans. M1 macrophages promote tissue inflammation and fibrosis by secreting TNF-α, IL-12, and iNOS. M2 macrophages, known for their anti-inflammatory effects, secrete IL-4 and IL-10, enhancing angiogenesis, lymphangiogenesis, and tissue healing through their migration [5,9,14,26]. Preclinical studies have linked M2 macrophage migration to vascular endothelial growth factor C (VEGF-C) expression, which regulates lymphangiogenesis at wound sites, making it a primary initiator of this process [5,27,28].

Active vitamin D_3_ (calcitriol) can enhance macrophage phenotype transition from M1 to M2, protecting renal function by preventing podocyte damage and glomerular dysfunction in diabetic rats and improving clinical manifestations via pathways such as STAT-1/TREM-1 or peroxisome proliferator-activated receptor-γ (PPAR-γ) [10,14].

The safe dose of calcitriol in rats ranges from 0.1 to 2 µg/kg/day [9,29]. In a previous study from our laboratory, calcitriol 1.2 μg/kg/day was administrated intraperitoneally, and the rats’ blood calcitriol levels were analyzed in order to evaluate the adverse effects of the drug [30]. According to their results this dose could be considered safe. Yet, there is no accurate data in the literature related to toxic doses of vitamin D in rats. In our study, the highest calcitriol consumption occurred in group B, which received the treatment both preoperatively and postoperatively. A daily dose of 1.2 µg/kg was administered to minimize possible side effects while maintaining an average dose consistent with the literature.

Histological analysis revealed that the operated right leg showed higher levels of collagen accumulation, macrophage migration, and LYVE-1-positive staining vessels than the non-operated left leg. Among all groups, group B exhibited significantly less collagen retention, lower M1 macrophage levels, higher M2 macrophage levels, higher M2/M1 macrophage balance, and new lymphatic micro-vessel formation. Conversely, group A, which served as the control and did not receive calcitriol, showed opposite results in these parameters (Figure 8).

Thus, calcitriol administration demonstrated a positive correlation with inflammation resolution, particularly in group B, which received the treatment both preoperatively and postoperatively. This group showed the most favorable outcomes in terms of micro-CT, fluorescence lymphatic imaging, and histological analyses, indicating that preoperative vitamin D administration significantly contributes to inflammation and lymphedema resolution. Group C demonstrated better histological and clinical outcomes than group A, whereas the results were less favorable than in group B. In fact, these findings support the use of preoperative and prophylactic active vitamin D.

Additionally, a positive correlation was observed between M2 macrophage migration and LVYE-1-positive newly formed micro-lymphatic vessels in lymphedematous tissues. However, it remains uncertain whether the formation of new lymphatic vessels is driven by VEGF-C release from migrated macrophages, antifibrinolytic effects of macrophages, or both.

Establishing a lasting lymphedema model in rats is known to be more difficult than in humans due to their extensive superficial lymphatic network [2,17]. Several factors may contribute to this phenomenon. First, rats have direct lymphatic vessel connections between the axillary and inguinal lymph nodes; this is a unique anatomical feature that couldn’t be found in other animal models or humans [31]. This may explain their ability to compensate for lymphatic drainage through collateral vessels, even when major lymphatic vessels are ligated or removed. Second, M2 macrophage migration to damaged tissues in rats is faster than in humans, potentially accelerating tissue regeneration, scar tissue formation, and fibrosis resolution.

Fibrosis is a common cause of end-organ failure in several systems, including the liver, lungs, skin, kidneys, and heart. Similarly, lymphedema may result from end-organ failure of the lymphatic system due to fibrosis. Macrophages play a crucial role in lymphangiogenesis and fibrosis, and understanding their molecular responses in lymphedematous tissues could improve our knowledge of lymphedema pathogenesis in humans. According to this study, vitamin D prophylaxis provided the best results on clinical and histological results of lymphedema. Calcitriol could be added to the regimen of patients who are prone to present lymphedema in the future. In addition, including calcitriol to treatment regimens may enhance therapeutic outcomes.

## 5. Conclusions

The superficial lymphatic network of rats was blocked by skin excision and radiotherapy, and active vitamin D was administered to study macrophage trafficking and its effects on fibrosis.

Among all groups, group B (prophylaxis group) exhibited significantly less collagen retention, lower M1 macrophage levels, higher M2 macrophage levels, higher M2/M1 macrophage balance, higher new lymphatic micro-vessel formation, and the most favorable outcomes in terms of micro-CT and fluorescence lymphatic imaging. Group A (control group) demonstrated the opposite results in these parameters. Group C (treatment group) illustrated better histological and clinical outcomes than group A, yet the results were less favorable than in group B. In fact, these findings support the use of preoperative and prophylactic active vitamin D.

## Figures and Tables

**Figure 1 medicina-60-01788-f001:**
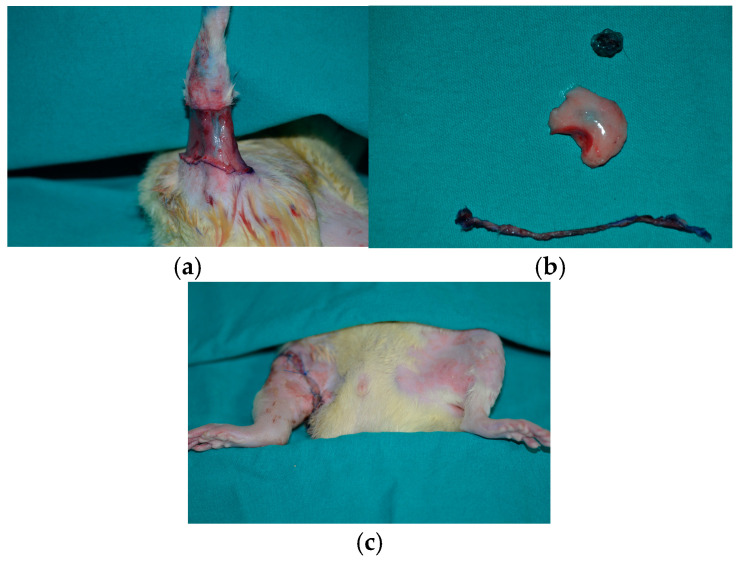
Surgical procedure. (**a**) Patent blue used to identify superficial lymphatic vessels as well as popliteal and inguinal lymph nodes. (**b**) The inguinal and popliteal lymph nodes, along with a 0.5-cm-wide circumferential strip of skin and subcutaneous tissue, excised from the right hind limb. (**c**) Postoperative third-week macroscopic appearance of the right and left hind limbs. The swollen right hind limb is visible (group C).

**Figure 2 medicina-60-01788-f002:**
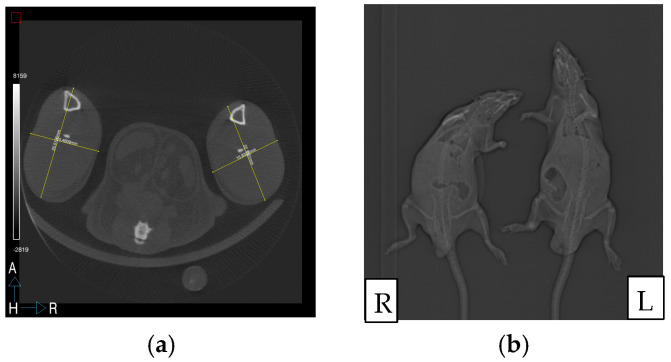
(**a**) Radiological analysis. The length and width of the soft tissue in the tibiofibular ankle measured and multiplied to determine the volume of the extremity in a rat from group B (**b**). Micro-computed tomography imaging of a rat from group A (L = left) and a rat from group C (R = right) in the eighth week of the study. Radiologic examination revealed lymphedema in the right hind limb of the rat from group A.

**Figure 3 medicina-60-01788-f003:**
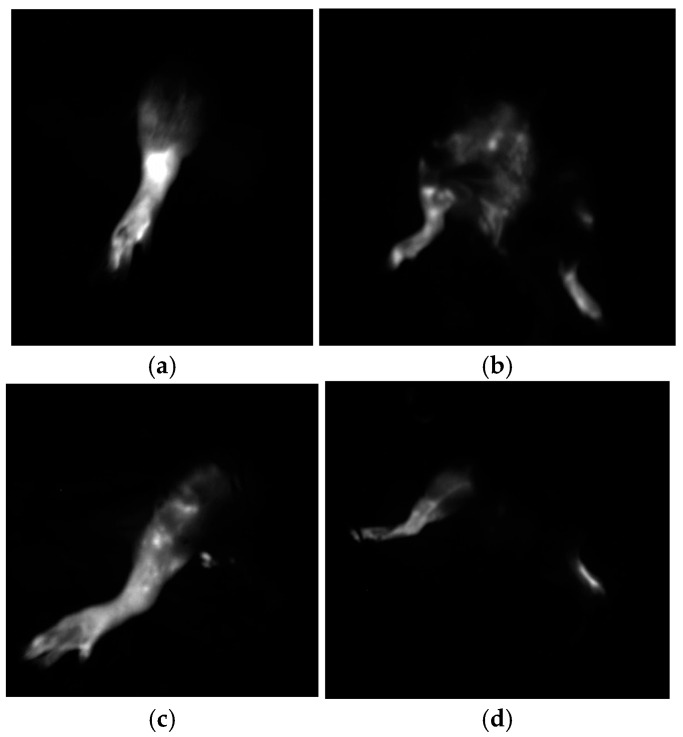
Fluorescence lymphatic imaging (**a**). Third-week fluorescence lymphatic imaging of a rat from group A (rat 6). The diffuse pattern is visible (**b**). Tenth-week fluorescence lymphatic imaging of a rat from group A (rat 6). ICG dye accumulation is visible at the distal part of the extremity (**c**). Third-week fluorescence lymphatic imaging of a rat from group B (rat 14). The splash-stardust pattern is visible. ICG transition proximal to the incision line is observed through the new superficial lymphatic network (**d**). Tenth-week fluorescence lymphatic imaging of a rat from group B (rat 14). The linear pattern and ICG dye passing are noticeable.

**Figure 4 medicina-60-01788-f004:**
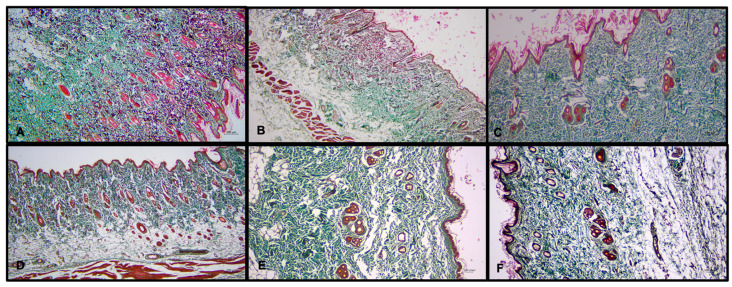
Masson’s trichrome staining for collagen accumulation. Increased fibroblastic proliferation in the deep dermis continues through the subcutaneous tissue and skeletal muscle, accompanying dense, thick collagen mass, mostly seen in group A. Group A: Right Leg (**A**); Group B: Right Leg (**B**); Group C: Right Leg (**C**); Group A: Left Leg (**D**); Group B: Left Leg (**E**); Group C: Left Leg (**F**).

**Figure 5 medicina-60-01788-f005:**
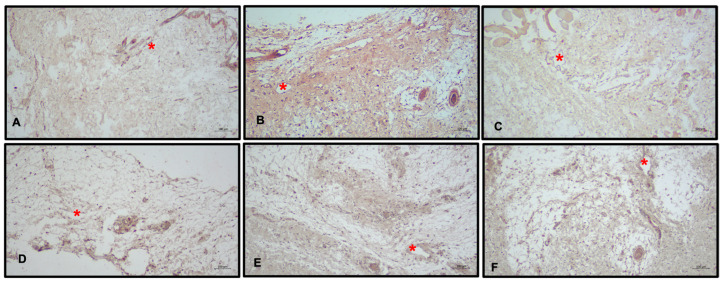
LYVE-1 immunohistochemical staining. Lymphatic vessel endothelial receptor 1-positive lymphatic vessels (asteriks) mostly observed in group B. Group A: Right Leg (**A**); Group B: Right Leg (**B**); Group C: Right Leg (**C**); Group A: Left Leg (**D**); Group B: Left Leg (**E**); Group C: Left Leg (**F**).

**Figure 6 medicina-60-01788-f006:**
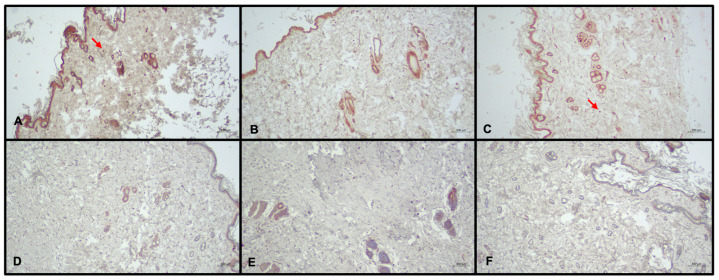
IL-12 immunohistochemical staining, arrow demonstrates M1 macrophages. Group A: Right Leg (**A**); Group B: Right Leg (**B**); Group C: Right Leg (**C**); Group A: Left Leg (**D**); Group B: Left Leg (**E**); Group C: Left Leg (**F**).

**Figure 7 medicina-60-01788-f007:**
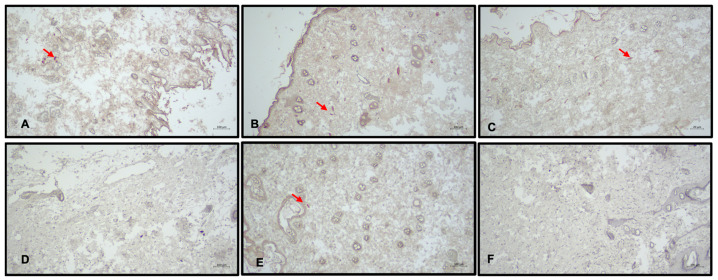
Arginase-1 immunohistochemical staining, arrow demonstrates M1 macrophages. Group A: Right Leg (**A**); Group B: Right Leg (**B**); Group C: Right Leg (**C**); Group A: Left Leg (**D**); Group B: Left Leg (**E**); Group C: Left Leg (**F**).

**Figure 8 medicina-60-01788-f008:**
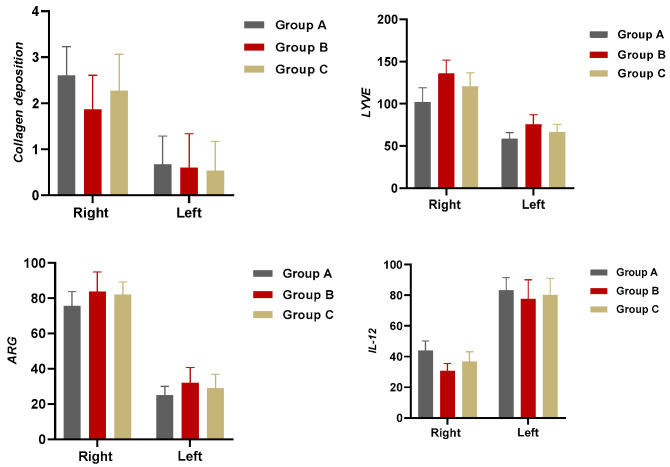
Column charts summarizing the histological results.

**Table 1 medicina-60-01788-t001:** Comparison of micro-computed tomography results among the groups.

		LYMPHEDEMA		
Groups	Total, *n*	4th Week Examination	8th Week Examination	*p*-Value
Group A	4	3 (75%)	3 (75%)	1 ^a^
Group B	4	2 (50%)	1 (25%)	0.999 ^a^
Group C	4	3 (75%)	1 (25%)	0.500 ^a^
*p*-value		0.999 ^b^	0.455 ^b^	

*p* > 0.05; a, McNemar Test; b, Fisher’s Exact Test.

**Table 2 medicina-60-01788-t002:** Comparison of histological staining results between the groups.

			Right	Left	Difference		
Variables	Groups	n	Mean ± SD	Mean ± SD	Mean [95%CI]	t	*p*-Value
Collagen Accumulation	Group A	15	2.60 ± 0.63	0.67 ± 0.62	1.93 [1.60–2.26]	**12.614**	**<0.001 ***
	Group B	15	1.87 ± 0.74	0.60 ± 0.74	1.27 [1.01–1.52]	**10.717**	**<0.001 ***
	Group C	15	2.27 ± 0.80	0.53 ± 0.64	1.73 [1.48–1.99]	**14.666**	**<0.001 ***
	**F**		**3.814**	0.150	**6.827**		
	** *p* ** **-value**		**0.030 ***	0.861	**0.003 ***		
	**Difference ****		*a > b*		*b < a*, *c*		
Lyve-1 Staining	Group A	15	102.33 ± 16.68	58.67 ± 7.19	43.67 [34.46–52.88]	**10.168**	**<0.001 ***
	Group B	15	136.33 ± 15.52	75.67 ± 11.47	60.67 [50.58–70.75]	**12.902**	**<0.001 ***
	Group C	15	120.67 ± 16.02	66.67 ± 9.00	54.00 [43.06–64.94]	**10.590**	**<0.001 ***
	**F**		**16.800**	**12.316**	**3.307**		
	***p*-value**		**<0.001 ***	**<0.001 ***	**0.046 ***		
	**Difference ****		*b > c > a*	*b > a*, *c*	*b > a*		
Arginase-1 Staining	Group A	15	75.60 ± 8.15	25.07 ± 4.95	50.53 [45.25–55.82]	**20.503**	**<0.001 ***
	Group B	15	83.73 ± 11.08	32.00 ± 8.59	51.73 [44.42–59.04]	**15.180**	**<0.001 ***
	Group C	15	82.00 ± 7.13	29.07 ± 7.81	52.93 [47.24–58.63]	**19.946**	**<0.001 ***
	**F**		**3.441**	**3.422**	0.175		
	***p*-value**		**0.041 ***	**0.042 ***	0.840		
	**Difference ****		*b > a*	*b > a*	NA		
IL-12 Staining	Group A	15	44.00 ± 6.23	83.20 ± 8.27	−39.20 [(−44.20)–(34.20)]	16.807	**<0.001 ***
	Group B	15	30.67 ± 4.82	77.73 ± 12.28	−47.07 [(−53.15)–(−40.99)]	16.606	**<0.001 ***
	Group C	15	36.80 ± 6.36	80.13 ± 10.89	−43.33 [(−51.52)–(−35.14)]	11.349	**<0.001 ***
	**F**		**19.544**	1.000	1.656		
	***p*-value**		**<0.001 ***	0.376	0.203		
	**Difference ****		*a > c > b*	NA	NA		

* *p* < 0.05, F = One-way ANOVA test, t = Paired sample *t*-test, SD = Standard deviation, CI = Confidence interval, ** = Tukey post hoc test, NA = Not available.

## Data Availability

The data presented in this study are available on request from the corresponding author.

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
