# Peer review of "Evaluation of the Effectiveness of Active Vitamin D Use in Experimental Rat Lymphedema Model"

_medicina, 2024, doi:10.3390/medicina60111788_

Round 1

Reviewer 1 Report

Comments and Suggestions for Authors

Dear Editor and Authors,

In the study, the authors found that active vitamin D calcitriol suppressed fibrosis and reduced lymphedema via M2 macrophage activation in rats in which lymphedema was induced with surgery and radiotherapy. This is an interesting study. However, there are many issues that need to be clarified, especially regarding the methodology of the study. After evaluating the study as comprehensively as possible, you can find the suggestions that I think will contribute to the development of the paper below. I wish the authors success in their work.

1. I suggest that the title of the study be changed to ‘Evaluation of the Effectiveness of Active Vitamin D Use in Experimental Rat Lymphedema Model’.

2. Radiotherapy is not mentioned in the method section of the abstract.

3. Rewrite the conclusion section of the abstract.

4. There is no Sham group in the study. How will the authors clarify this situation?

5. It seems more logical and understandable to call Group B of the study groups as the prophylaxis group and Group C as the treatment group.

6. By what method were the rats sacrificed at the end of the study?

7. Provide a reference for the anesthesia doses used in the study.

8. What type of radiation was used in the study? X-ray radiation?

9. The radiation dose used in the study, although limited in area, appears to be a very high dose. The rats were followed alive for a relatively long period of 10 weeks after this dose. Did any rats die before the study was completed? Did the rats show signs of radiation toxicity? I am concerned about the welfare of the animals. Provide more references for the radiation dose used.

10. In which laboratory was the study conducted? Which guidelines were followed throughout the study? Or was it not conducted according to any guidelines?

11. Provide information on the manufacturer's name, catalog number, and drug form of calcitriol used in the study. What form of calcitriol was used? If oral, how was it administered intraperitoneally? In what substance was calcitriol dissolved? Oral forms cannot be administered sterile, and animals almost certainly develop peritonitis when administered intraperitoneally. Also, why was this particular dose of calcitriol chosen? Provide a valid reason and reference for the selected application times and period of calcitriol. No reference is provided in the method section. Although it is partially mentioned in the discussion section. Also, could the authors have done this study with 25OHD vitamins and compared both forms?

12. Please provide a figure showing the study design.

13. Provide more information about the manufacturer, catalogue number, and dilution ratios for the IHC kits.

14. Explain with references how many slides and areas were evaluated from how many rats in total for histological and immunohistochemical analyses. Also provide references on the method by which you evaluated these semiquantitative analyses. Also provide additional information on how the assessment was performed. For example, when evaluating the positivity rates, did you determine an area and calculate the percentage of positivity in that area? This is very unclear, and your method needs to be explained. If I want to evaluate these analyses by referencing this study, I cannot find how to do this in this paper.

15. The ages of the rats used in the study are not specified. Because the weights seem a bit high to me.

16. What method did you use to determine the number of animals in the groups in the study? Please provide a reference for your method. For example, could you have used fewer or more animals? Why are there 15 animals in each group, not 12 or 16? Provide more information about this.

17. Why were male rats specifically chosen for the study? Can we apply the results to females? How do gender-specific differences affect these results? Please clarify this issue by discussing it in the discussion section.

18. Provide positive and negative controls for the IHC analyses used in the study. In addition, biochemical studies of several other cytokines secreted by M1 and M2 macrophages may increase the reliability of the results. If possible, provide this as well.

19. The most important problem is that we cannot know whether any adverse effects occurred due to calcitriol administration in this study. Because calcium, phosphorus, vitamin D, and creatine levels were not studied in rats after calcitriol administration. Therefore, we have no idea about the safety of the applied dose. How do the authors plan to overcome this problem?

20. In the Discussion section, provide clinical insights and future perspectives regarding the results of the study.

21. Please paraphrase the Conclusion section.

Comments on the Quality of English Language

Minor editing in English is needed.

Author Response

Reviewer: 1

Dear Reviewer please find our replies according to your valuable comments below.

Comments to the Author

  1. I suggest that the title of the study be changed to ‘Evaluation of the Effectiveness of Active Vitamin D Use in Experimental Rat Lymphedema Model’.

Answer: We aggre with you. The title has been changed.

(Page 1, line 1-3)

             “Evaluation of the Effectiveness of Active Vitamin D Use in Experimental Rat Lymphedema  Model”

  1. Radiotherapy is not mentioned in the method section of the abstract. Answer: The information is added into abstract section.                                                                 

Abstract/ Material and Methods (page 1, line 47-48)

One week after the surgical ablation a total dose of 20 Gy radiation therapy was administered to the operated groin region.

  1. Rewrite the conclusion section of the abstract.

Answer: The conclusion has been changed.

Abstract/ Conclusion (Page 2, line 58-63)

Conclusion: Both histological analysis and clinical results reported a relevant influence of Calcitriol administration. Among all groups, the most favorable outcomes were seen in Group B (prophylaxis group) Hence, calcitriol administration could play a crucial role in enhancing the migration of M2 macrophages to the damaged tissue, and such migration may contribute to lymphedema resolution either by enhancing the organization of superficial lymphatic vessels or resolving fibrosis, or with a combination of both these mechanisms.”

  1. There is no Sham group in the study. How will the authors clarify this situation?

Answer: Creating long lasting lymphedema is difficult in rats than humans due to their advanced regeneration capacity. We thought that in sham group we would not create even an acute lymphedema because of rats healing capacity. We would not obtain accurate data for the influence of vitamin D to macrophage migration since their wound healing mechanism is fast and there would not be lymphedema after sham surgery. Therefore, we applied to the surgical ablation and radiotheraphy to our control group in order to create long lasting lymphedema.

  1. It seems more logical and understandable to call Group B of the study groups as the prophylaxis group and Group C as the treatment group.

Answer: The changes have been made.

Material and Methods/ Lymphedema Rat Model (page 3, line 105-108)

The groups were as follows: control group (group A), prophylaxis vitamin D group (group B; preoperative 7 days and postoperative 10 weeks of 1.2 μg/kg per day intraperitoneal calcitriol), and treatment vitamin D group (group C; postoperative 10 weeks of 1.2 μg/kg per day intraperitoneal calcitriol).

  1. By what method were the rats sacrificed at the end of the study?

Material and Methods/ Histological Analysis  (Page 4, line 159-161)

Answer All rats were sacrificed in the 10th week after surgery. All rats were euthanized with high dose sodium pentobarbital (Pentobarsol, Dechra Veterinary Products, USA) (150–200 mg/ml). [9].

The reference is also added into manuscript.

“Zhou KL, Zhang YH, Lin DS, Tao XY, Xu HZ. Effects of calcitriol on random skin flap survival in rats. Sci Rep. 2016;6:18945. Epub 20160106. doi: 10.1038/srep18945. PubMed PMID: 26732750; PubMed Central PMCID: PMCPMC4702169”

  1. Provide a reference for the anesthesia doses used in the study.

Answer: The reference is added into manuscript.

Material and Methods/ Lymphedema Rat Model (Page 3, Line 111-113)

General anesthesia was administered with intraperitoneal injections of 87.5-mg/kg ketamine hydrochloride (Pfizer, New York) and 12.5-mg/kg xylazine (Rompun; Bayer Healthcare)[2],

“Aksoyler D, Bitik O, Menku Ozdemir FD, Gokoz O, Uzun H, Yeniceri B, et al. A New Experimental Lymphedema Model: Reevaluating the Efficacy of Rat Models and Their Clinical Translation for Chronic Lymphedema Studies. Ann Plast Surg. 2021;86(6):707-13. doi: 10.1097/SAP.0000000000002479. PubMed PMID: 32759623.”

  1. What type of radiation was used in the study? X-ray radiation?

Answer: X-ray radiation was used.

Material and Methods Lymphedema Rat Model (Page 3, line 124-129)

“One week after the surgical procedure, under general anesthesia with the same agents and doses,      a single dose of radiation was applied to a 4 × 4-cm area over the operated groin region at a rate of 1 Gy/min with a total dose of 20 Gy (Varian 2100 EX Linear Accelerator; Medical Imaging Resources, Ann Arbor, Mich). Topical hydrocortisone cream was applied to any radiation-related burns [16,17]

  1. The radiation dose used in the study, although limited in area, appears to be a very high dose. The rats were followed alive for a relatively long period of 10 weeks after this dose. Did any rats die before the study was completed? Did the rats show signs of radiation toxicity? I am concerned about the welfare of the animals. Provide more references for the radiation dose used.

Answer: Kanter at all. reported 9 months rat survival time after 45 Gy single dose administration. We preferred 20 GY single dose of radiation in accordance with literature. The related references are also included into manuscript. After the radiotheraphy topical hydrocortizon was also applied.

Material and Methods/ Lymphedema Rat Model (Page 3, line 124-129)

One week after the surgical procedure, under general anesthesia with the same agents and doses, a single dose of radiation was applied to a 4 × 4-cm area over the operated groin region at a rate of 1 Gy/min with a total dose of 20 Gy (Varian 2100 EX Linear Accelerator; Medical Imaging Resources, Ann Arbor, Mich). Topical hydrocortisone cream was applied to any radiation-related burns [16,17]

“Yang CY, Nguyen DH, Wu CW, Fang YH, Chao KT, Patel KM, et al. Developing a Lower Limb Lymphedema Animal Model with Combined Lymphadenectomy and Low-dose Radiation. Plast Reconstr Surg Glob Open. 2014;2(3):e121. Epub 20140407. doi: 10.1097/GOX.0000000000000064. PubMed PMID: 25289315; PubMed Central PMCID: PMCPMC4174147.

“Harb AA, Levi MA, Corvi JJ, Nicolas CF, Zheng Y, Chaudhary KR, et al. Creation of a Rat Lower Limb Lymphedema Model. Ann Plast Surg. 2020;85(S1 Suppl 1):S129-S34. doi: 10.1097/SAP.0000000000002323. PubMed PMID: 32205493.”

Discussion 1st paragraph (Page 9, line 309-312)

“We used the rat lymphedema model described by Kanter et al. [1]. This model involved circumferential skin and subcutaneous tissue excision, inguinal and popliteal lymphadenectomy, followed by a single dose (20 Gy) of radiotherapy. The radiotherapy aimed to block the rats' strong superficial lymphatic collateral circulation and observe macrophage subtype responses to fibrosis [2].

“Kanter MA, Slavin SA, Kaplan W. An experimental model for chronic lymphedema. Plast Reconstr Surg. 1990;85(4):573-80. doi: 10.1097/00006534-199004000-00012. PubMed PMID: 2315397.”

  1. In which laboratory was the study conducted? Which guidelines were followed throughout the study? Or was it not conducted according to any guidelines?

Answer: The relevant information was added into text.

Material and Methods/ Lymphedema Rat Model (Page 3, line 101-103)

All animal procedures were performed and complied with Animal Research Center of Aziz Sancar Experimental Medicine Research Institue Laboratory Animal Production and Purification Department Guidelines.

Page 12, line 422-424

Institutional Review Board Statement: This animal study was conducted with the approval of the Istanbul University Animal Experiments Local Ethics Committee, Istanbul, Turkey (Approval No. 2023/13).

  1. Provide information on the manufacturer's name, catalog number, and drug form of calcitriol used in the study. What form of calcitriol was used? If oral, how was it administered intraperitoneally? In what substance was calcitriol dissolved? Oral forms cannot be administered sterile, and animals almost certainly develop peritonitis when administered intraperitoneally. Also, why was this particular dose of calcitriol chosen? Provide a valid reason and reference for the selected application times and period of calcitriol. No reference is provided in the method section. Although it is partially mentioned in the discussion section. Also, could the authors have done this study with 25OHD vitamins and compared both forms?

Answer:  Calcitriol (Calcijex (1mcg/1 ml) solution for injection ampoules  Abbott Laboratories, Chicago, IL, USA) was used for the study. Injectable form of solution was applied intraperiteonally. It was already in dissolved form. We did not confront any peritonitis or related issue in study. The safe dose of calcitriol in rats ranges from 0.1 to 2 µg/kg/day. A previous study from our laboratory administrated 1.2 μg/kg/day intraperitoneally and analyzed the rats’ blood calcitriol levels in order to scrutinize the adverse effects of the drug. According to their results this dose could be considered safer level of the calcitriol. Therefore, 1.2 μg/kg/day was applied in this current study. This reference was added into discussion section and the drug and manufacturer info with its refernce added into methods section. In literature the researches related to calcitriol administration in rats described daily dose of calcitriol. Initially, we also tried to find half life of calcitriol in rats in Pubmed however we could not obtain any specific information about this. In literature we only find the articles about the active form of vitamin D. Therefore, we conducted our study based on active vitamin D administration.

Material and Methods/ Lymphedema Rat Model (Page 3, line 108-109)

Calcitriol (Calcijex (1mcg/1 ml) solution for injection ampoules Abbott Laboratories, Chicago, IL, USA) was used in this study[15].

de Almeida LF, Francescato HDC, da Silva CGA, Costa RS, Coimbra TM. Calcitriol reduces kidney development disorders in rats provoked by losartan administration during lactation. Sci Rep. 2017;7(1):11472. Epub 20170913. doi: 10.1038/s41598-017-11815-8. PubMed PMID: 28904363; PubMed Central PMCID: PMCPMC5597607.

Discussion page 10, line 340-344

The safe dose of calcitriol in rats ranges from 0.1 to 2 µg/kg/day [9,29]. In a previous study from our laboratory, calcitriol 1.2 μg/kg/day was administrated intraperitoneally and rats’ blood calcitriol levels was analyzed in order to evaluate the adverse effects of the drug[30]. According to their results this dose could be considered safe. Yet, there is not any accurate data in literature related to toxic dose of vitamin D in rats. 

Yucel HC, Yalcin Y, Akpinar OF, Cayli M, Ozdemir I, Solakoglu S, et al. Effectiveness of 1alpha-25-dihydroxyvitamin D3 active substance on anastomosis safety in the rat femoral artery end-to-end anastomosis experimental model: Macroscopic and histological analyses. J Plast Reconstr Aesthet Surg. 2024;97:310-9. Epub 20240814. doi: 10.1016/j.bjps.2024.08.049. PubMed PMID: 39213932.

  1. Please provide a figure showing the study design.

Answer: Dear reviewer, although we tried to explain the study step by step with current images we totally understand that an illustration would make the study design more understandable. Unfortunately, due to economic issues we cannot afford the illustration fee of the study design. We are so sorry for this omission.

  1. Provide more information about the manufacturer, catalogue number and ratios for the IHC kits.

Answer: Material and Method/ Histological Analyses (Page 4, line 170-175)

lymphatic vessel endothelial receptor 1 (LYVE-1; Abcam, Inc., Cambridge, Massachusetts, Catalog#AB 281587), M1 macrophages stained with IL-12 antibody (Anti-IL12A Antibody; Picoband, Boster Biological Technology, Pleasanton CA, USA, Catalog # A00918-1), and M2 macrophages stained with Arginase-1 antibody (Anti-IL12A Antibody; Abcam, Inc., Cambridge, Massachusetts, Catalog# A00918-1)

  1. Explain with references how many slides and areas were evaluated from how many rats in total for histological and immunohistochemical analyses. Also provide references on the method by which you evaluated these analyses. Also provide additional information on how the assessment was performed. For example, when evaluating the positivity rates, did you determine an area and calculate the percentage of positivity in that area? This is very unclear, and your method needs to be explained. If I want to evaluate these analyses by referencing this study, I cannot find how to do this in this paper.

Answer:

Material and Method/ Histological Analyses (Page 4 line 162-163)

Fifteen rats from each group were sampled. Skin samples were harvested from a point 4 mm distal to the surgical wound

Material and Method/ Histological Analyses (Page 4, Line 175-185)

Arginin 1 and Il-12 staining sections were evaluated semiquantitatively. Cell density in the tissue samples was examined by two different histologists and scored between 0 and 3. If there were no stained cells, it was scored as “0”, if there were a small number of stained cells in isolation, it was scored as “1”, if there were stained cells forming small clusters, it was scored as “2” and if the stained cells formed large clusters, it was scores as “3”. One slide from each subject was stained and ten fields were evaluated on each slide. H-score values were determined by calculating the average scores in the histological scoring formula. H-score is obtained by the formula which is 3x% of strongly staining cells (3+intensity)+2x% of moderately staining cells (2+intensity)+1x% of weakly staining cells (1+intensity), in the range of 0-300.[18] The observers were blinded to the lymphatic vessels and macrophages on specimens.

Ou, D.; Adam, J.; Garberis, I.; Blanchard, P.; Nguyen, F.; Levy, A.; Casiraghi, O.; Gorphe, P.; Breuskin, I.; Janot, F.; et al. Influence of tumor-associated macrophages and HLA class I expression according to HPV status in head and neck cancer patients receiving chemo/bioradiotherapy. Radiother Oncol 2019, 130, 89-96, doi:10.1016/j.radonc.2018.08.013.

  1. The ages of the rats used in the study are not specified. Because the weights seem a bit high to me.

Answer: 8-month-old rats were used in this study.

Material and Methods/ Lymphedema Rat Model (Page 3, line 99-100)

Forty-five non-immunosuppressed male 8-month old Sprague–Dawley rats, weighing between 350 and 400 g, were randomly separated into three groups (= 15 per group).

  1. What method did you use to determine the number of animals in the groups in the study? Please provide a reference for your method. For example, could you have used fewer or more animals? Why are there 15 animals in each group, not 12 or 16? Provide more information about this.

Answer:

Material and Methods/ Statistical Analysis (Page 5, line 186-195)

The calculation (d-value) method developed by Cohen was used to calculate the effect size to be used as a reference in determining the sample size.[19] The effect size for the least significant measurement difference was calculated as d = 0.81, using the findings of the study in the literature investigating the new lymphedema model on the rats.[2] In this context, in the analysis of the quantitative difference between the two groups, d = 0.81 for a change difference of approximately 0.9 units (±1.1 standard deviations); It was deemed appropriate to include a total of 45 rats as the sample group using the G-power (version 3.1.9.7, Kiel, Germany) package program with a 95% confidence level (1-α) and 90% test power (1-β).

Cohen, J. A power primer. Psychol Bull 1992, 112, 155-159, doi:10.1037//0033-2909.112.1.155.

Aksoyler, D.; Bitik, O.; Menku Ozdemir, F.D.; Gokoz, O.; Uzun, H.; Yeniceri, B.; Nasir, S.N. A New Experimental Lymphedema Model: Reevaluating the Efficacy of Rat Models and Their Clinical Translation for Chronic Lymphedema Studies. Ann Plast Surg 2021, 86, 707-713, doi:10.1097/SAP.0000000000002479.

  1. Why were male rats specifically chosen for the study? Can we apply the results to females? How do gender-specific differences affect these results? Please clarify this issue by discussing it in the discussion section.

Answer: In our instution, female rats are usually provided for reproductive system related studies.

Discussion 2nd paragraph (Page 10, Line 314-316)

Lymphedema progression in humans is not gender dependent. Even though male rats were chosen in this study in accordance with literature, results could be implemented to female rats.[2,16]

Aksoyler D, Bitik O, Menku Ozdemir FD, Gokoz O, Uzun H, Yeniceri B, et al. A New Experimental Lymphedema Model: Reevaluating the Efficacy of Rat Models and Their Clinical Translation for Chronic Lymphedema Studies. Ann Plast Surg. 2021;86(6):707-13. doi: 10.1097/SAP.0000000000002479. PubMed PMID: 32759623.

Yang CY, Nguyen DH, Wu CW, Fang YH, Chao KT, Patel KM, et al. Developing a Lower Limb Lymphedema Animal Model with Combined Lymphadenectomy and Low-dose Radiation. Plast Reconstr Surg Glob Open. 2014;2(3):e121. Epub 20140407. doi: 10.1097/GOX.0000000000000064. PubMed PMID: 25289315; PubMed Central PMCID: PMCPMC4174147.

  1. 18. Provide positive and negative controls for the IHC analyses used in the study. In addition, biochemical studies of several other cytokines secreted by M1 and M2 macrophages may increase the reliability of the results. If possible, provide this as well.

Answer:

Results/ Histopathological Analyses (Page 7, Line 251-255)

The positive (Right Leg) and negative controls (Left Leg) for the IHC analyses can be seen in the Table 2. Several different antibodies can be used. However, we used IL 12 antibody for M1 macrophage staining and Arginase-1 antibody for M2 macrophage staining because their accuracy and reliability has already been confirmed by similar studies.

Ref

Zhang X, Zhao Y, Zhu X, Guo Y, Yang Y, Jiang Y, et al. Active vitamin D regulates macrophage M1/M2 phenotypes via the STAT-1-TREM-1 pathway in diabetic nephropathy. J Cell Physiol. 2019;234(5):6917-26. Epub 20181127. doi: 10.1002/jcp.27450. PubMed PMID: 30478987.

Ghanta S, Cuzzone DA, Torrisi JS, Albano NJ, Joseph WJ, Savetsky IL, et al. Regulation of inflammation and fibrosis by macrophages in lymphedema. Am J Physiol Heart Circ Physiol. 2015;308(9):H1065-77. Epub 20150227. doi: 10.1152/ajpheart.00598.2014. PubMed PMID: 25724493; PubMed Central PMCID: PMCPMC4551121.

Zhang X, Zhou M, Guo Y, Song Z, Liu B. 1,25-Dihydroxyvitamin D(3) Promotes High Glucose-Induced M1 Macrophage Switching to M2 via the VDR-PPARgamma Signaling Pathway. Biomed Res Int. 2015;2015:157834. Epub 20150419. doi: 10.1155/2015/157834. PubMed PMID: 25961000; PubMed Central PMCID: PMCPMC4417570. Zhang XL, Guo YF, Song ZX, Zhou M. Vitamin D prevents podocyte injury via regulation of macrophage M1/M2 phenotype in diabetic nephropathy rats. Endocrinology. 2014;155(12):4939-50. Epub 20140904. doi: 10.1210/en.2014-1020. PubMed PMID: 25188527.

  1. The most important problem is that we cannot know whether any adverse effects occurred due to calcitriol administration in this study. Because calcium, phosphorus, vitamin D, and creatine levels were not studied in rats after calcitriol administration. Therefore, we have no idea about the safety of the applied dose. How do the authors plan to overcome this problem?

Answer: One of study from our instution researched the effects of calcitriol on the femoral artery anastomoses patency in the rats. In their preliminary work demonstrated that the 1.2 μg/kg/day intraperitoneal calcitriol administration did not cause any harm in the rats. During the their experiment, blood samples also were taken from rats in order to calculate the vitamin D blood levels. Yet, there is not any accurate data in literature related to toxic dose of vitamin D in rats. In this dose, there were not any metabolic adverse effects seen. Therefore, we applied the same dose.

Discussion 7th parapraph (Page 10, Line 340-344)

The safe dose of calcitriol in rats ranges from 0.1 to 2 µg/kg/day [9,29]. In a previous study from our laboratory, calcitriol 1.2 μg/kg/day was administrated intraperitoneally and rats’ blood calcitriol levels was analyzed in order to evaluate the adverse effects of the drug[30]. According to their results this dose could be considered safe. Yet, there is not any accurate data in literature related to toxic dose of vitamin D in rats.

Yucel, H.C.; Yalcin, Y.; Akpinar, O.F.; Cayli, M.; Ozdemir, I.; Solakoglu, S.; Demiroz, A.; Aksoyler, D.Y. Effectiveness of 1alpha-25-dihydroxyvitamin D3 active substance on anastomosis safety in the rat femoral artery end-to-end anastomosis experimental model: Macroscopic and histological analyses. J Plast Reconstr Aesthet Surg 2024, 97, 310-319, doi:10.1016/j.bjps.2024.08.049.

  1. In the Discussion section, provide clinical insights and future perspectives regarding the results of the study.

Discussion last paragraph- Page 11, line 390-399

Answer: Fibrosis is a common cause of end-organ failure in several systems, including the liver, lungs, skin, kidneys, and heart. Similarly, lymphedema may result from end-organ failure of the lymphatic system due to fibrosis. Macrophages play a crucial role in lymphangiogenesis and fibrosis, and understanding their molecular responses in lymphedematous tissues could improve our knowledge of lymphedema pathogenesis in humans. According to these study vitamin D prophylaxis provided best results on clinical and histological results of lymphedema. Calcitriol could be added to the regimen of patients who are prone to present lymphedema in future. In addition, including calcitriol to treatment regimens may enhance therapeutic outcomes.

  1. Please paraphrase the Conclusion section.

Answer: Conclusion is modified.

Conclusion second paragraph, page 12, line 405-412,

Among all groups, group B (prophlaxy group) exhibited significantly less collagen retention, lower M1 macrophage levels, higher M2 macrophage levels, higher M2/M1 macrophage balance, higher new lymphatic microvessel formation and most favorable outcomes in terms of micro-CT, fluorescence lymphatic imaging. Group A (control group) demonstrated the opposite results in these parameters. Group C (treatment group) illustrated better histological and clinical outcomes than Group A yet the results were less favorable than Group B.  In fact, these findings support the use of preoperative and prophylactic active vitamin D.

Thank you very much for your precious comments.

Reviewer 2 Report

Comments and Suggestions for Authors

As a rule, the development of secondary lymphedema of the lower extremities is based on the endothelial dysfunction. According to statistics, lymphedema is diagnosed in one out of 10 patients who come to the doctor with complaints of leg edema. The main reasons for the development of secondary lymphedema include acquired organic and functional disorders of lymph flow. Complications developing on the background of lower limbs lymphedema led to impaired mobility, self-care of patients, disability and reduce their quality of life. Therefore, the search for new, economically feasible, effective and at the same time uncomplicated ways of treatment of lymphatic drainage disorders is urgent. Until now there are no unambiguous answers in science for the formation of a single theory of pathogenesis of this pathology. But according to the available data different pathogenetic factors playing a role in the development of lymphedema are singled out, although their priority is estimated controversially. The relevance of studying the course of pathogenetic mechanisms in lower limb lymphedema on the background of vitamin D support therapy reveals for us new approaches to the principles of pathogenetic treatment, which, considering some causes of its development (after surgical operations, during radiation therapy, etc.) may become preventive.

In the present study, results were obtained that confirm the positive effects of vitamin D administration in the perioperative period in rodents. These results may form the basis for appropriate supportive therapy in patients with identified risks of lower limb lymphedema. The work is well illustrated, the data reflected in the tables confirm the need for clinical studies regarding the effect of different doses of vitamin D on the prevention and levelling of pathogenetic mechanisms leading to the development of lymphedema, which is important for the formation of algorithms of effective pathogenetic correction and improvement of the quality of life of patients.

The authors of the paper have some minor comments and questions:

1. Figure 2. The figure should be within the fields, preferably larger in size. Full caption for the figure should be on the same page where it is located. Please, correct it. Label which group the CT scan is from in Figure 2 a. In figure 2c, labelled L or R.

2. Figure 3, Table 3: place also in the fields of the text.

3. Figure 3, in brackets: do you indicate the number assigned to the rat in the group or the number of rats from the group? If the number, you should write 6 rats, not the rat 6, etc.

4. Table 2 is on 2 pages, please, write the word ‘Continuation of Table 2’ on the second page.

5. Figure 4: To improve the perception of the information, I suggest a slight change in the figure caption at the end.

Group A: Right Leg (A); Group B: Right Leg (B); Group C: Right Leg (C);

               Left Leg    (D);                Left Leg   (E);                 Left Leg    (F).

6. The same (item 5) is for figures 5-7.

7. You use "M1/M2 macrophage ratio" in your paper, but there is no direct data anywhere on their change in ratio. Either add the data or change the interpretation of the data in the paper.

8. In the conclusion you should add the results (data) obtained to the phrases already written.

9. References are not arranged according to the rules of the journal. Correct it.

Best regards.

Author Response

Reviewer: 2

Dear Reviewer please find our replies according to your valuable comments below.

Comments to the Author

  1. Figure 2. The figure should be within the fields, preferably larger in size. Full caption for the figure should be on the same page where it is located. Please, correct it. Label which group the CT scan is from in Figure 2 a. In figure 2c, labelled L or R.

Answer: The figure size is changed. Caption and the figure are inserted in the same page and CT scan of a rat from group A is also mentioned. In figure 2b, label L and R are inserted the side of figure 2b.

Results/ Limb Volume Calculation (Page 5, Line 218-224)

  1. Figure 3, Table 3: place also in the fields of the text.

Answer:  The change has been made.

Figure 3: Results/ Fluorescence lymphatic imaging  (Page 6, Line 238-247)

Table 3: Discussion (Page 11, line 362-366)

  1. Figure 3, in brackets: do you indicate the number assigned to the rat in the group or the number of rats from the group? If the number, you should write 6 rats, not the rat 6, etc.Answer:

Answer:  We indicated the number assigned to the rat in the group. Rat 6 indicates sixth rat from group A.

  1. Table 2 is on 2 pages, please, write the word ‘Continuation of Table 2’ on the second page.

Answer: The table 2 is fit in a single page.

Results/ Histopathological Analyses (Page 7, Line 251-255)

  1. Figure 4: To improve the perception of the information, I suggest a slight change in the figure caption at the end.

Group A: Right Leg (A); Group B: Right Leg (B); Group C: Right Leg (C);

               Left Leg    (D);                Left Leg   (E);                 Left Leg    (F).

Answer: The change has been made according to your suggestion.

Results/ Histopathological Analyses/ Collagen accumulation (Page 8, Line 271-272)

  1. The same (item 5) is for figures 5-7.

Answer: The changes have been made according to your suggestion.

Figure 5: Results/ Histopathological Analyses/ LYVE-1 Staining (Page 8, Line 283-284)

Figure 6: Results/ Histopathological Analyses/ IL-12 Staining (Page 9, Line 293-294)

Figure 7: Results/ Histopathological Analyses/ Arginase-1 Staining (Page 9, Line 303-304)

  1. You use "M1/M2 macrophage ratio" in your paper, but there is no direct data anywhere on their change in ratio. Either add the data or change the interpretation of the data in the paper.

Answer: What we actually meant by the M1/M2 macrophage ratio was the M1/M2 macrophage balance. Therefore, ratio word is changed to balance throughout the paper. 

Abstract/ Background (Page 1, Line 42)

Keywords (Page 2, Line 65)

Introduction 3th paragraph (Page 2, Line 89)

Discussion 7th paragraph (Page 10, Line 352-353)

Conclusion 2nd paragraph (Page 12, Line 407)

  1. In the conclusion you should add the results (data) obtained to the phrases already written.

Answer: Conclusion is modified.

Conclusion second paragraph, page 12, line 405-412,

Among all groups, group B (prophlaxy group) exhibited significantly less collagen retention, lower M1 macrophage levels, higher M2 macrophage levels, higher M2/M1 macrophage balance, higher new lymphatic microvessel formation and most favorable outcomes in terms of micro-CT, fluorescence lymphatic imaging. Group A (control group) demonstrated the opposite results in these parameters. Group C (treatment group) illustrated better histological and clinical outcomes than Group A yet the results were less favorable than Group B.  In fact, these findings support the use of preoperative and prophylactic active vitamin D.

  1. References are not arranged according to the rules of the journal. Correct it.

Answer: References has been corrected.

Thank you very much for your precious comments.

Round 2

Reviewer 1 Report

Comments and Suggestions for Authors

The authors have mostly addressed the comments and provided convincing answers. However, it would have been nice if they had included the IHC dilution rates. I wish the authors success in their work.

Author Response

Dear Reviewer

Thank you very much for your precious comments. 

Comments

The authors have mostly addressed the comments and provided convincing answers. However, it would have been nice if they had included the IHC dilution rates. I wish the authors success in their work.

Answer: 

Material- Methods /Histological analysis- 2nd paragraph  Page 4 (Line 175-176)

  The dilution ratios for LVYE-1, IL-12 and Arginin- 1 antibodies for immunohistochemistry (IHC) were 1:100, 1:200 and 1:200, respectively